# Synthesis and Structure of Novel Copper(II) Complexes with N,O- or N,N-Donors as Radical Scavengers and a Functional Model of the Active Sites in Metalloenzymes

**DOI:** 10.3390/ijms22147286

**Published:** 2021-07-06

**Authors:** Joanna Masternak, Małgorzata Zienkiewicz-Machnik, Iwona Łakomska, Maciej Hodorowicz, Katarzyna Kazimierczuk, Milena Nosek, Amelia Majkowska-Młynarczyk, Joanna Wietrzyk, Barbara Barszcz

**Affiliations:** 1Institute of Chemistry, Jan Kochanowski University in Kielce, Uniwersytecka 7, 25-406 Kielce, Poland; barbara.barszcz@ujk.edu.pl; 2Institute of Physical Chemistry, Polish Academy of Sciences, Kasprzaka 44/52, 01-224 Warsaw, Poland; 3Faculty of Chemistry, Nicolaus Copernicus University in Toruń, Gagarina 7, 87-100 Toruń, Poland; 4Faculty of Chemistry, Jagiellonian University, Ingardena 3, 30-060 Kraków, Poland; hodorowm@chemia.uj.edu.pl; 5Department of Chemistry, Gdańsk University of Technology, Gabriela Narutowicza 11/12, 80-233 Gdańsk, Poland; katarzyna.kazimierczuk@pg.edu.pl; 6Faculty of Rehabilitation, University of Rehabilitation in Warsaw, Kasprzaka 49, 01-234 Warsaw, Poland; milena.dietetyk@gmail.com; 7Endocrinology Clinic, Regional Hospital in Kielce, Artwińskiego 6, 25-734 Kielce, Poland; amelia.m@op.pl; 8Hirszfeld Institute of Immunology and Experimental Therapy, Polish Academy of Sciences, R. Weigl 12, 53-114 Wrocław, Poland; joanna.wietrzyk@hirszfeld.pl

**Keywords:** copper(II) complexes, N,O- and N,N-donors, X-ray crystal structure, antioxidant activity, enzyme mimetic

## Abstract

To evaluate the antioxidant activity of potential synthetic enzyme mimetics, we prepared new five copper(II) complexes via a self-assembly method and named them [Cu(2-(HOCH_2_)py)_3_](ClO_4_)_2_ (**1**), [Cu(2-(HOCH_2_)py)_2_(H_2_O)_2_]SiF_6_ (**2**), [Cu_2_(2-(HOCH_2_CH_2_)py)_2_(2-(OCH_2_CH_2_)py)_2_](ClO_4_)_2_ (**3**), [Cu(pyBIm)_3_](BF_4_)_2_·1.5H_2_O (**4**) and [Cu(py_2_C(OH)_2_)_2_](ClO_4_)_2_ (**5**). The synthetic protocol involved N,O- or N,N-donors: 2-(hydroxymethyl)pyridine (2-(HOCH_2_)py), 2-(hydroxyethyl)pyridine (2-(HOCH_2_CH_2_)py), 2-(2-pyridyl)benzimidazole (pyBIm), di(2-pyridyl)ketone (py_2_CO). The obtained Cu(II) complexes were fully characterised by elemental analysis, FTIR, EPR, UV-Vis, single-crystal X-ray diffraction and Hirshfeld surface analysis. Crystallographic and spectroscopic analyses confirmed chromophores of both monomeric ({CuN_3_O_3_} (**1**), {CuN_2_O_4_} (**2**), {CuN_6_} (**4**), {CuN_4_O_2_} (**5**)) and dimeric complex ({CuN_2_O_3_} (**3**)). Most of the obtained species possessed a distorted octahedral environment, except dimer **3**, which consisted of two copper centres with square pyramidal geometries. The water-soluble compounds (**1**, **3** and **5**) were selected for biological testing. The results of the study revealed that complex **1** in solutions displayed better radical scavenging activity than complexes **3**, **5** and free ligands. Therefore, complex **1** has been selected for further studies to test its activity as an enzyme mimetic. The chosen compound was tested on the erythrocyte lysate of two groups of patients after undergoing chemotherapy and chemoradiotherapy. The effect of the tested compound (**1**) on enzyme activity levels (TAS, SOD and CAT) suggests that the selected complex can be treated as a functional mimetic of the enzymes.

## 1. Introduction

The content of this publication is a continuation of research conducted by our group [1,2,3], whose studies are focused on the search for new transition metal coordination compounds with selected small-molecule heteroaromatic ligands. Our research allows us to contribute to solving important problems of contemporary science related to medicine, health care and environmental conditions. Due to the contemporary state of environmental pollution in which humans live, research into the metabolic role of oxygen has increased in recent years. Specifically, an oxygen molecule can undergo both a complete, four-electron reduction to a water molecule (the process that forms the basis of intracellular respiration) and a gradual, one-electron reduction, resulting in the formation of reactive oxygen species (ROS). Reactive species can be divided into two groups: (i) ROS that are free radicals, such as superoxide anion radicals (O_2_˙^−^), hydroperoxide radicals (HO_2_˙) or hydroxyl radicals (OH˙), and (ii) ROS that do not have an unpaired electron, such as singlet oxygen (^1^O_2_), (^1^Δg) ozone (O_3_) or hydrogen peroxide (H_2_O_2_). ROS are characterised by a very high reactivity, as they react with almost all cellular components and, thus, can cause damage to all molecular classes of various cell components [4,5]. The metabolic consequences of this damage affect the body, playing a significant role in the pathogenesis of many diseases, such as neurodegenerative diseases [6,7,8] and cancer [9], and can lead to inborn metabolic abnormalities of red blood cells and the development of anaemia [10]. It is important to note that increased levels of reactive oxygen species, which lead to serious organ damage, are also a consequence of therapies, including chemo- and radiotherapy [11] and reperfusion after ischaemic conditions, e.g., in transplantation.

Based on the above-mentioned facts, it should be clearly emphasised that the basis for the proper functioning of the organism is the balance between ROS and the body’s antioxidant barrier. Ongoing studies have shown that due to the high level of environmental pollution, natural antioxidants (antioxidant enzymes such as catalase, glutathione peroxidase, superoxide dismutase or vitamin C, uric acid, glutathione vitamin E, carotenoids and ubihydroquinone) are not always able to provide an effective defence of the organism against oxidative stress [12]. Hence, the search is being made for small-molecule combinations—metalloenzyme mimetics is carried out in parallel. This is all the more important because attempts to use natural enzyme preparations for therapeutic purposes have shown unsatisfactory pharmacodynamic and pharmacokinetic results in the treatment process. The reason is the enzymes’ high molecular weight and charge density, which hinders their penetration through cell membranes. Thus, the synthetic antioxidants can support cancer treatment, especially with chemotherapy and radiotherapy, as they can be a valuable way to achieve the correct antioxidant balance in the body.

Therefore, in our lab, we synthesised three antioxidants containing Mn(II) ions in their centres and characterised them by Mn-CAT activity in aqueous solutions [13,14]. The complexes belong to monomeric ([Mn(NCS)_2_(2-(CH_2_)_2_OHpy)_2_]) [13], dimeric ([Mn_2_(μ-Cl)_2_(2-CH_2_OHpy)_4_]Cl_2_·2H_2_O) [13] and polymeric ([Mn(SO_4_)(H_2_O)(2-CH_2_OHpy)]_n_) systems [14]. Pyridine derivatives serve as heteroatomic ligands; these include 2-(hydroxymethyl)pyridine (2-CH_2_OH), and 2-(hydroxyethyl)pyridine (2-(CH_2_)_2_OHpy), which coordinate as N,O-donors. We tested structurally characterised complexes with pyridine derivatives as catalysts for the H_2_O_2_ decomposition reaction in an aqueous solution [13,14]. The oxygen evolution over time was monitored using a quadrupole mass spectrometer (QMS). The accompanying changes in the electron structure of the metal during the oxidation-reduction reaction of manganese ions (Mn(II) ⇆ Mn(III)) were observed in situ using resonance inelastic X-ray scattering (RXES) spectroscopy. The recorded two-dimensional RXES maps confirmed the involvement of Mn ions in oxidation and reduction processes. The [Mn_2_(*μ*-Cl)_2_(2-CH_2_OHpy)_4_]Cl_2_·2H_2_O complex (TOF = 1.87·10^−2^ min^−1^) was found to have the highest catalytic activity among the studied mimetics [13]. Moreover, the obtained complexes were water-soluble, which enabled their penetration through cell membranes and, thus, good uptake by the organism.

Continuing the research on synthetic antioxidants, the main goal of the current work is the synthesis and physicochemical characterisation of new complexes with copper(II) central ions as potential enzyme mimetics (SOD, CAT). The study’s complementary aim was to evaluate the antioxidant activity of the obtained complexes and then to test the best antioxidant as a mimetic of SOD and CAT enzymes in blood samples (erythrocyte lysate) of patients after chemotherapy and chemoradiotherapy. To achieve these goals, we obtained five new copper(II) complexes and fully characterised them by elemental analysis, single-crystal X-ray diffraction, FTIR, UV-Vis and EPR spectroscopy. Moreover, Hirshfeld surface analysis was used to verify the contributions of the different intermolecular interactions. The potential antioxidant effects of water-soluble complexes **1**, **3** and **5** were determined via ABTS assay. The Cu(II) complex (**1**) that exhibited the best radical-scavenging activity was used as an enzyme mimetic in the next step of our experiments. The final step was to evaluate the effect of a selected copper(II) complex on the level of antioxidant activity in a group of patients after chemotherapy and chemoradiotherapy using the RANDOX test for TAS level, the RANSOD test (Randox Laboratories, Crumlin, UK) for SOD activity, as well as CAT and GPx assay kits (Calbiochem, Merck, Darmstadt, Germany).

## 2. Results and Discussion

### 2.1. Synthetic Considerations in the Self-Assembly of Copper(II) Complexes

Cu(II) is classified as an intermediate class of Lewis acids according to the Hard and Soft Acid and Bases model, which suggested that this ion would form stable bonds with similar, i.e., belonging to the borderline class of Lewis bases. Therefore, for this research, we chose N,O- and N,N- donor ligands: 2-(hydroxymethyl)pyridine, 2-(hydroxyethyl)pyridine, 2-(2-pyridyl)benzimidazole and di(2-pyridyl)ketone—for the synthesis of stable copper(II) complexes (Scheme 1 and Scheme 2). Such selected ligands are polydentate and can form stable chelate rings with a metal ion due to free electron pairs of the endocyclic nitrogen atoms, which are very good nucleophiles and provide binding sites with the central ion. Among the selected ligands, the most interesting compound in terms of polydenticity is di(2-pyridyl)ketone (py_2_CO), which has three potential donor atoms: two nitrogen atoms of two pyridine rings and a carbonyl oxygen atom. Therefore, it can adopt different coordination patterns, for example: monodentate (κN), chelating (κN,N′) and bridging-chelating (*μ*-κ^2^N,O:O,N′). Furthermore, as we noticed during the synthesis of Cu(II) complexes, in the presence of water, the hydrolysis of the organic ligand py_2_CO to the gem-diol py_2_C(OH)_2_ [15] also took place and led to the complex [Cu(py_2_C(OH)_2_)_2_](ClO_4_)_2_ (**5**) formation. Scheme 1 illustrates the synthetic routes and proposed structures of the copper(II) complexes derived from Cu(ClO_4_)_2_ salt.

In order to check the role of the anion in the Cu(II) complexation process in our synthesis, the second source of copper ions was Cu(BF_4_)_2_·H_2_O salt. The copper complexes obtained according to published procedures are shown in Scheme 2.

It should be emphasised that in the presence of traces of water due to the hydrolysis reaction of tetrafluoroborate anions, the [Cu(2-(HOCH_2_)py)_2_(H_2_O)_2_]SiF_6_ (**2**) complex was unexpectedly obtained. The counterion SiF_6_^2−^ was formed in accordance with the following equations [16]:$$BF_{4}^{-}+4H_{2}O\;\to {\left[B{\left(OH\right)}_{4}\right]}^{-}+4HF$$
$$4HF+SiO_{2\left(glass\right)}\;\to \;SiF_{4}+2H_{2}O$$
$$2SiF_{4}+2H_{2}O\to \;H_{2}{\mathrm{SiF}}_{6}+SiO_{2}+2\mathrm{HF}\;$$

### 2.2. Physicochemical Characterisation of the Complexes

#### 2.2.1. Structural Studies

The crystallographic data and detailed information on the structure solution and refinement for all copper complexes are provided in Table 1.

##### Molecular Structure of Complex **1**

In the structure of compound **1**, copper(II) ion is coordinated by 2-(hydroxymethyl)pyridine molecules in a chelating manner via pyridine nitrogen atom and oxygen atom from the hydroxymethyl group (Figure 1a). Thus, the central copper(II) ion is in an N_3_O_3_ coordination environment arranged in a distorted octahedral geometry, as evidenced by both bond distances and all of the angles deviating (see Appendix A). The basal plane around copper is achieved by three nitrogen atoms with almost the same bond distances [Cu(1)-N(1) 2.023(3), Cu(1)-N(2) 2.037(4) and Cu(1)-N(3) 2.021(3) Å] and with an oxygen atom [Cu(1)-O(3) 1.939(2) Å], whose bond is shorter, as mentioned above. The bond lengths in axial positions equal to Cu(1)-O(1) 2.295(3) Å and Cu(1)-O(2) 2.426(3) Å are lengthened. Moreover, the tetragonality parameter T = 0.85 (T = R_S_/R_L_; where R_S_ means Cu-L equatorial bond length and R_L_ means Cu-L axial bond length) [17], indicates static tetragonal distortion, as a consequence of the Jahn–Teller effect [18,19]. The two perchlorate anions (one of which is disordered and shows the arrangement of a distorted tetragonal bipyramid) balanced the charge of the copper(II) centre. Moreover, the hydroxyl groups of 2-(HOCH_2_)py ligand participate in conventional O-H∙∙∙O/Cl hydrogen bonds, as summarised in Figure 1b and Appendix A, which were confirmed additionally by HS analysis (Appendix A).

##### Molecular Structure of Complex **2**

The corresponding complex fragment [Cu(2-(HOCH_2_)py)_2_(H_2_O)_2_]^2+^ (**2**) consists of octahedrally coordinated Cu(II) ion with two bidentate 2-(hydroxymethyl)pyridine ligands and two water molecules (see Figure 2a and Appendix A). The copper(II) ion is located 0.004 Å above the plane constructed by Cu(1)-O(8) 2.014(1) Å, Cu(1)-O(17) 2.001(1) Å Cu(1)-N(1) 1.990(1) Å and Cu(1)-N(9) 1.997(1) Å. It is noteworthy that the octahedron around the copper ion is elongated in the axial direction, exhibiting a Jahn–Teller effect (Cu(1)-O(16) 2.240(1) Å and Cu(1)-O(18) 2.313(1) Å). The value of the T parameter, indicating the degree of tetragonal elongation of the octahedron in **2**, is equal to 0.87. Nevertheless, the Cu-O and Cu-N bond distances are similar to the analogous bond found in the literature for copper(II) complexes containing 2-(hydroxymethyl)pyridine [20,21,22,23].

The overall charge +2 on the metal centre is counterbalanced by the hexafluorosilicate anion generated in situ. The slight distortion from the ideal octahedral geometry of SiF_6_^2-^ arising from a deviation in Si-F distances and angles is attributed to the network of hydrogen bonds and interactions. All the contacts listed in Appendix A are mostly to fluorine ions with bond lengths of approximately 2.7 Å, as observed by the O∙∙∙F distance in O-H∙∙∙F hydrogen bonds [24], and approximately 3.3 Å for the C∙∙∙F distance of the C-H∙∙∙F hydrogen bonds (Figure 2b). The crystal packing also reveals intra- and intermolecular hydrogen bonds. Accordingly, the intramolecular hydrogen bond involves the H(2) atom of the pyridine ring and O(17) atom of the water molecule, while the intermolecular C(13)-H(13)∙∙∙O(18)_(x, −y + ½, z + ½)_ hydrogen bond also engages the pyridine ring and second water molecule.

##### Molecular Structure of Complex **3**

The molecular structure of **3** consists of dinuclear cationic units and two perchlorate anions (Figure 3a). Each Cu(II) centres is coordinated by (2-hydroxyethyl)pyridine in a chelating manner (κN,O) and by pyridyl-2-ethanolato ions expressing chelating and bridging modes of coordination (*μ*_2_-κ^2^N,O:κO). Thus, the oxygen atoms (O(2) and O(2)^i^) of the deprotonated hydroxyl group are simultaneously linked to two metal ions. As a consequence, the bond distance of the oxygen atom of the bridged group is shorter (1.923(5)–1.936(5) Å) than that of the remaining Cu-O(1) bond (2.288(6) Å). The Cu-N distances are in the typical range of 1.992(7)–2.024(6) Å (Appendix A). Furthermore, within the dimer, the Cu⋯Cu distance is equal to 3.0286(17) Å. The copper ion is five-coordinated with a N_2_O_3_-type chromophore. Hence, the basal plane of square pyramidal coordination geometry (τ = 0.09) [25] is formed by two pyridine-N donor atoms and two deprotonated pyridyl-2-ethanolato-O atoms.

The crystal structure of **3** is stabilised by a series of intra- and intermolecular C-H···O interactions (Appendix A). The intramolecular interactions involved the oxygen atoms of the hydroxyethyl group of chelating 2-(HOCH_2_CH_2_)py (Figure 3b). Moreover, the oxygen atoms of the non-coordinated perchlorate ions are involved in a series of intermolecular interactions with the hydrogen atoms of the neighbouring pyridine molecules.

##### Molecular Structure of Complex **4**

Complex **4** with the molecular formula [Cu(pyBIm)_3_](BF_4_)_2_·1.5H_2_O consists of copper(II) ions coordinated by three pyBIm molecules, BF_4_^−^ counterions and waters of crystallisation. The water molecule (O(1)) exhibits a fractionally occupied (50%) position. The copper(II) ion shows six coordination bonds with N-donor atoms of the chelating pyBIm ligand (Figure 4a) forming a distorted octahedron. The abovementioned Cu-N distances in the range of 1.909–2.252 Å are consistent with those observed in related compounds [26]. One of the pyBIm ligands, unlike the other two, is positionally disordered about the 2-fold axis. This type of disorder was also observed in the previously reported structure of the Cu(II) complex [27]. Additionally, the disorder can also be noticed in the case of the tetrafluoroborate groups. The BF_4_^−^ ions are involved in a bifurcated intramolecular H-bonding interaction with one water molecule of crystallisation (see Figure 4b, Appendix A). In addition, intramolecular interactions (C-H···N) involving the N-pyridine group and benzene rings stabilised the packing of the molecules in the crystal.

##### Molecular Structure of Complex **5**

Among the ligands used by us in the synthesis of copper(II) complexes, the di-2-pyridyl ketone (py_2_CO) stands out due to its polydenticity, as it has three potential donor atoms (N, O, N′). In addition, in the presence of water and metal ions, the carbonyl group undergoes a process of a water molecule addition to form *gem*-diol (py_2_C(OH)_2_). The structure of **5** consists of a [Cu(py_2_C(OH)_2_)_2_]^2+^ cation, containing two *gem*-diols, and ClO_4_^−^ counterions (Figure 5a). The Cu(II) ion exhibits a Jahn–Teller-distorted octahedral geometry with tridentate κ^3^N,O,N′-coordination of the py_2_C(OH)_2_ ligand. The axial Cu-O bond length (2.4429(1) Å) is longer than the equatorial distances Cu-N (2.0083(19)–2.0097(19) Å), which are comparable to previously reported copper complexes [28,29]. The crystal structure of **5** is stabilised by series of hydrogen bonds in the range of 2.787–3.5938 Å, mainly between the non-coordinated -OH groups of the *gem*-diol and the ClO_4_^−^ ions (Appendix A and Figure 5b).

According to the literature [28,29,30,31,32,33,34,35,36,37,38], it appears that py_2_CO can exhibit different coordination modes in the presence of Cu(II) ions; it is most often found in the *gem*-diol form, as the neutral ligand py_2_C(OH)_2_, or, less frequently, as the anion py_2_CO(OH)^−^. It should be noted that the anionic form of the ligand favours the formation of multicore copper(II) complexes. Namely, in the dimer [Cu_2_Br_3_(C_11_H_9_N_2_O_2_)] [30], the ligand plays a bridging role linking the two metal centres via a deprotonated oxygen atom, which is illustrated by the coordination model *μ*-κ^4^N,O:O,N′. Interestingly, an analogous coordination mode occurs in the tetramer [Cu_4_[(2-Py)_2_CO(OH)]_2_(O_2_CCH_3_)_6_(H_2_O)_2_]·CH_2_Cl_2_ [34]. Based on the data collected in Table 2, it should be noted that the most common coordination mode of *gem*-diol in copper(II) complexes is κ^3^N,O,N′. The complex obtained by B. L. Westcott et al. [38] is also noteworthy; in this complex, the ligand behaves as an N,N′-donor (coordination model κ^2^N,N′), and the oxygen atoms of the hydroxyl groups are not involved in the coordination. Based on the collected literature data, it is also possible to observe whether the structure of the complexes is affected by the type of used salt anions (source of Cu(II) ions). The collected data indicated that Br^−^ ions could play the dual role of both bridging ligands and counterions. A similar phenomenon was observed for complexes with CH_3_COO^−^ ions [32,34,37]. Additionally, copper complexes with *gem*-diol should be divided into (i) monomeric complexes with a distorted octahedral coordination centre [28,29,31,32,33] and (ii) dimeric complexes with a polyhedron in the form of a distorted square pyramid [30,38].

#### 2.2.2. Hirshfeld Surface Analysis of the Complexes

The interaction of low molecular weight molecules with biological media may play a vital role in the mimetic activity. One purpose in the present study is to show the multifaceted relationship between not only the structural type of the complexes but also the opportunity to form noncovalent interactions in biological systems. To achieve this goal, it necessary was to use the Hirshfeld surface analysis method (HS). The Hirshfeld surface visualisations with fingerprint plots of the noncovalent interactions of all complexes are presented in Appendix A. The corresponding contribution percentages of the major interactions shown in Figure 6 suggested that the analysed crystal structures differ significantly. In particular, the percentage of O···H/H···O interactions that appear as long and asymmetrical spikes in the fingerprint plots of all complexes reflects a notable difference. All complexes with -OH groups and ClO_4_^−^ counterions display the O···H/H···O interactions in the range of 30.3–44.9%, while the proportion of these interactions makes up 5.3 or 5.8% of the HS for complexes with water molecules and SiF_6_^2−^ or BF_4_^−^ counterions. The main type of interaction in complexes **3** and **5** is that of H···F/F···H close contacts, which contributes between 32.2 and 20.8% of the HS, respectively. In addition, the symmetrical wings indicate C···H/H···C contacts and participation of C-H···π interactions between alkyl groups and/or aromatic fragments. The complementary regions located in the centre of fingerprint plots for all complexes corresponding to C···C interactions are significantly smaller (1.2–3.1%), which may be due to the presence of counterions in the crystal lattice preventing interactions between aromatic fragments of the complexes. The second prevailing type of interactions in all studied copper(II) complexes are H···H contacts, with their contribution ranging from 24.9% to 46.7% for **5** and **3**, respectively.

#### 2.2.3. FTIR Spectra

To study the binding mode of N,O- (2-(HOCH_2_)py, 2-(HOCH_2_CH_2_)py, py_2_CO) or N,N-donor (pyBIm) ligands to the copper ion in the analysed complexes, the IR spectrum of the free ligand was compared with the spectra of the complexes (see Appendix A). Due to the simultaneous ν(O-H) stretching of the alcohol groups (**1**, **2**, **3** and **5**) and water molecules (in the case of complexes **2** and **4**), the IR spectra in the high-frequency region have broad and weak absorption bands centred at 3425 (**1**), 3381 (**2**), 3450 (**3**), 3590 (**4**) and 3485 cm^−1^ (**5**). To confirm engagement of the free electron pairs of the endocyclic nitrogen atoms of the pyridine ring and oxygen atom of the hydroxyl group, IR spectroscopic delta values (Δν = ν_complex_ − ν_ligand_) were calculated and are listed in Table 3. The results indicated that the ligands used are very good nucleophiles and provide binding sites with the central ion. Furthermore, in the case of complexes **1**, **3** and **5**, peaks at 1091 (**1**), 1086 (**3**) and 1090 cm^−1^ (**5**) can be assigned to the ν_as_(Cl-O) band of ClO_4_^−^ [39]. The FTIR spectrum of compound **2** with SiF_6_^2−^ counterions exhibits the bands at 764 and 725 cm^−1^ [40]. Moreover, it should be noted that the stretching vibration of the BF_4_^−^ ions in the FTIR spectrum of **4** was assigned to the band at 1052 cm^−1^ [39].

#### 2.2.4. UV/Visible Spectra

The electronic spectral analysis in solution was based on assumed octahedral symmetry with expected single bands corresponding to the ^2^T_2g_ ← ^2^Eg transition, which is usual for a regular octahedron [41]. However, the bands observed for many complexes are very broad and decidedly asymmetric due to the Jahn–Teller effect, indicating the presence of a lower (D_4h_) symmetry, as expected for tetragonally distorted complexes.

The Cu(II) complex **1** studied here exhibits a d–d absorption band (Figure 7a, Table 4), in the 550–700 nm region, in accordance with its coordination cores {CuN_3_O_3_} [42]. Thus, the ethanol absorption spectrum of **1** shows a single broad absorption band with maxima equal to 654 nm, characteristic of a distorted octahedral structure. In the 400–200 nm region, at least two types of strong transitions associated with π→π* or n→π* can be connected with the intraorganic ligand (LL) [43]. However, in this spectrum, we did not observe batho- or hypsochromic shift associated with coordination. Similarly, the electronic spectrum (in methanol) of compounds 3–5 (Figure 7, Table 4) shows two strong bands below 300 nm due to ligand-centred n→π∗/π→π∗ (LLCT) transitions. Additionally, moderate broad or shoulder bands at 303 and 346 nm (**3**) are assigned to the ligand–metal (LMCT) charge transfer transitions. The visible spectra of **3** in the solution (Figure 7, Table 4) are similar to those of the species discussed above. This fact indicated that a five-coordinated sphere is surrounded by solvent molecules to make the CN equal to 6, and the maximum of d-d band occurs at 662 nm. More complicated d-d absorption spectra are observed for complexes **4** and **5** due to the Jahn–Teller effect. The spectral image shows a much broader band (λ_max_ 560 nm) of **5** {CuN_4_O_2_}, and the asymmetry increases the band (λ_max_ 702 nm) of **4** {CuN_6_}.

#### 2.2.5. Electron Paramagnetic Resonance Spectra and Magnetic Moment Measurement

One of the important research methods for the physicochemical characterisation of copper(II) complexes is EPR: the most direct and powerful method for the detection and identification of metal complexes with unpaired electrons and free radicals. The interpretation of the electron paramagnetic resonance spectrum can provide confirmatory information on the magnetic susceptibility of the studied complex samples. The results of structural studies on monocrystals presented above showed that complexes [Cu(2-(HOCH_2_)py)_3_](ClO_4_)_2_ (**1**), [Cu(pyBIm)_3_](BF_4_)_2_·1.5H_2_O (**4**) and [Cu(py_2_C(OH)_2_)_2_](ClO_4_)_2_ (**5**) are monomers with the structure of distorted octahedra. The configuration of the central ion 3d^9^ indicates the presence of one unpaired electron, which was confirmed by the values of effective magnetic moments measured at room temperature: 1.69, 1.56 and 1.52 BM, respectively. The magnetic data correlate with the results of the interpretation of EPR spectra exemplarily performed for complex **1**.

The spectra for monomeric complex **1** were recorded at room temperature and LNT in powder form as well as in frozen H_2_O solutions at 77 K (Figure 8). The EPR spectra of the polycrystalline powder **1** at both temperatures are very similar and dominated the slightly narrow signals typical for Cu(II) ions (S = 1/2). The estimated g_||_ and g_⊥_ values (2.19 and 2.05) for the complex display the order g_∥_ > g_⊥_ > 2.0023, which is consistent with a dx^2^-y^2^ ground state, and indicate the axially elongated octahedral mononuclear copper(II) complex due to Jahn–Teller distortion. Additionally, the EPR spectra recorded at 77 K in water (**1**) show four well-defined hyperfine lines due to coupling with copper nuclei in the parallel region. Such a spectrum image also confirmed the axial pattern of complex species in solution (Figure 8c). The EPR data for dimer **3** indicated the structural difference between solid-state and solution. The magnetic and EPR data obtained for the dimer [Cu_2_(2-(HOCH_2_CH_2_)py)_2_(2-(OCH_2_CH_2_)py)_2_](ClO_4_)_2_ (**3**) (powder form) indicated that this compound is EPR-silent and, thus, diamagnetic at room temperature. Neither full-field nor half-field signals were detected at room or liquid-nitrogen temperatures, and no signals of mononuclear Cu(II) impurities were detected. Diamagnetism and EPR silent **3** indicated a strong antiferromagnetic interaction between individual copper(II) ions. The same phenomenon was observed by Driessen et al. [44,45] for Cu(II) complexes with 1-(2-hydroxyethyl)-3,5-dimethylpyrazole. The exchange interaction was characteristically dependent on the bridge angle, implying a shift from ferromagnetic to antiferromagnetic coupling at approximately 97.6° [46]. With increasing bond angles, the magnetic orbitals of both Cu(II) ions are favourably oriented and delocalised towards the bridging network, which leads to very strong antiferromagnetic interactions. Comparing X-ray data, we noticed that for the dimer **3**, the Cu···Cu distance was equal to 3.0288(17) Å and the Cu-O-Cu angle was 103.37°; these results are very similar to those obtained by Driessen et al. [44,45] for dinuclear diamagnetic Cu(II) complexes. Unexpectedly, the EPR spectra of dimer in frozen solution indicate the presence of monomeric copper compound in the solution (Figure 9).

Such a phenomenon indicates a process of solvent-induced dimer breakdown of the Cu-O-Cu bonds. This observation is comparable to that reported by Repich et al. [47]. The EPR data for complex **3** in frozen methanol solution are as follows: g_||_ 2.33, g_⊥_ 2.08, A_||_ 154 G.

### 2.3. Biological Activity Research

The new structurally and spectroscopically characterised complexes were subjected to biological studies to evaluate their application as synthetic mimetics of enzymes based on copper(II). In the first stage, the Cu(II) complexes were evaluated for their antioxidant activity. Analysis of the obtained data allowed the selection of the most active free radical scavenger. Moreover, our study aimed to visualise the antioxidant status of patients after two types of treatment: chemotherapy and chemoradiotherapy. The final step was to evaluate the effect of a selected copper(II) complex on the level of antioxidant activity in a group of patients after chemotherapy and chemoradiotherapy.

#### 2.3.1. Free Radical Scavenging Ability of Copper(II) Complexes with Heteroaromatic Alcohols

Among the synthetic compounds used for modelling biological systems, mimetics that are soluble in solvents of neutral pH are particularly favoured because they can penetrate cell membranes and, thus, be absorbed by the organism. Therefore, among the coordination compounds isolated during the syntheses, those soluble in water (**1**, **3** and **5**) were selected as model systems for studies on antioxidant properties. Additionally, we checked the stability of selected complexes in buffer solution (PBS) at two different time points (0 min and 12 h). UV-Vis spectra confirmed the stability of the complexes at room temperature (Appendix A).

The antioxidant activity of the samples was evaluated by applying an ABTS radical assay test, which was chosen as a widely used nonenzymatic method to provide basic information on the capability of compounds to scavenge free radicals. The complexes with hydroxymethyl- and ethylpyridine in solution displayed higher radical scavenging activity (IC_50_ 0.26 ± 0.03 (**1**) and 0.62 ± 0.05 mM (**3**)) than free ligands (IC_50_ = 3.79 ± 0.23 mM (2-(HOCH_2_)py) and 7 mM (2-(HOCH_2_CH_2_)py)) (Figure 10). This feature can be attributed to the chelation of heteroaromatic alcohols with copper(II) ions to produce a strong synergistic effect for efficient radical scavenging. Hydroxymethylpyridine in complex **1** coordinates to form five-membered chelate rings, while hydroxyethylpyridine in complex **3** coordinates to six-membered rings. This fact probably differentiates the antioxidant properties of the above complexes. However, analysis of the experimental data also shows that the complex [Cu(py_2_C(OH)_2_)_2_](ClO_4_)_2_ (**5**) expresses the lowest antioxidant activity, and its IC_50_ is equal to 24.49 ± 0.50 mM. Surprisingly, given the IC_50_ of the free ligand (2.40 mM), there is no synergistic effect on the antioxidant activity of the ligand-complex system (**5**). Therefore, taking into consideration the IC_50_ for further studies, complex **1** (0.26 ± 0.03 mM) was selected. The chosen complex **1** was also tested for in vitro cytotoxicity against the malignant cell lines (A549, HT29, MCF-7) and healthy mouse fibroblasts (BALB/3T3). The reference chemotherapeutic drug was used cisplatin (IC_50_ 3.03 ± 0.2, 9.23 ± 0.77, 8.20 ± 0.48, 3.33 ± 0.38 μM, respectively). The results obtained have revealed anti-proliferative effects of complex **1** on cancerous cells (IC_50_ values at the level of 94.74 ± 3.9, 98.46 ± 2.7, 91.66 ± 2.9 μM and 106.48 ± 4.2, respectively). In fact, this compound exhibited about 30 times lower cytotoxicity against normal cell line compared to cisplatin.

#### 2.3.2. Activity Levels of Blood Antioxidants after Treatment of Oncology Patients and in the Control Group (Healthy Patients)

First, in cooperation with the Regional Hospital in Kielce, we evaluated the antioxidant status of twenty patients after they received treatment for breast cancer—women after mastectomy and chemotherapy or chemoradiotherapy and a healthy control group. Total antioxidant status (TAS) and SOD, CAT and GPx activity in all groups were evaluated using calorimetric assays. The results are expressed as the mean ± SEM. Table 5 presents a comparison of all the measured parameters in the studied groups.

By analysing the graphs in Figure 11, it could be easily noticed that the total antioxidant status (TAS) of the group of patients after chemotherapy was lower (1.05 ± 0.16) than that of the control group (1.44 ± 0.33). However, the SOD, in contrast to the TAS and CAT, was higher in the oncology patients than in the control group. Similar findings of SOD activity levels for patients with breast cancer were obtained by Tsai et al. [48] and Kasapović et al. [49] who suggested that the activity of antioxidative enzymes strictly depends on the type of cancer and its stage. Additionally, an increase in SOD activity may indicate increased defence activity of the body and may suggest a recurrence of cancer [50]. This may be because the SOD enzyme is the body’s first line of defence against the formation of ROS and protects against their reaction with cellular components. Superoxide dismutase catalyses the disproportionation reaction of superoxide anion radicals:2O_2_^•−^ + 2H^+^ → H_2_O_2_ + O_2_.

Additionally, the data presented in Table 4 indicate that the pattern of changes in GPx activity is analogous to that discussed above for the CAT enzyme.

Considering that the consequence of cancer therapies is an increase in reactive oxygen species, we wanted to investigate the antioxidant enzyme levels not only for a group of patients after chemotherapy but also for those who had undergone chemoradiotherapy. The results of the study presented in Figure 12 below showed the same direction of change in TAS, SOD and CAT levels in the group of healthy controls as was observed in the group of patients that had undergone chemotherapy. Nevertheless, it should be noted that the SOD levels of those who had undergone chemoradiotherapy were much higher (263.70 ± 126.87) than those of patients who had only received chemotherapy (221.13 ± 54.51). It is likely that such SOD activity correlates with the level of ROS after radiotherapy treatment [50].

#### 2.3.3. Evaluation of the Effect of a Selected Copper(II) Complex on the Level of Antioxidant Activity in a Group of Patients after Chemo- and Radiotherapy

One of the aims of this work was to synthesise enzyme-mimetics copper(II) complexes that effectively neutralise excess free radicals. Literature data indicate that structural and functional models can be distinguished among mimetics [51]. The copper(II) complex [Cu(2-(HOCH_2_)py)_3_](ClO_4_)_2_ (**1**), which expressed the best antioxidant activity in the ABTS assay (IC_50_ 0.26 ± 0.03), was selected for this study and was considered as a functional model of antioxidant enzyme mimetics. This compound, due to the presence of a metal ion with redox properties, can act primarily as a mimetic of the SOD enzyme, which catalyses the superoxide anion radical dismutation reaction according to the redox equations [52,53]:Cu^2+^ + O_2_^•−^ → Cu^+^ + O_2_
Cu^+^ + O_2_^•−^ + 2H^+^ → Cu^2+^ + H_2_O_2_

The next step was the neutralisation of hydrogen peroxide using a test mimetic that undertakes CAT functions [54]:2Cu^2+^ + H_2_O_2_ → 2Cu^+^ + O_2_ + 2H^+^
2Cu^+^ + H_2_O_2_ + 2H^+^ → 2Cu^2+^ + 2 H_2_O

Among the literature proposals of this type of mimetic [55,56], the selected copper(II) complex **1** deserves to be highlighted due to its good solubility in water. This feature facilitates the transport of the compound through cell membranes and, thus, increases the organism’s absorption process.

The antioxidant activity of enzyme mimetic **1** was tested on the blood of two groups of patients after (1) chemotherapy and (2) chemoradiotherapy. The effect of a tested mimetic [Cu(2-(HOCH_2_)py)_3_](ClO_4_)_2_ (**1**) on enzyme activity levels (TAS, SOD and CAT) in a group of patients after chemotherapy in comparison with antioxidant activity in control samples is shown in Figure 13. The addition of Cu(II) mimetic (25 µg/mL) to blood samples of patients after chemotherapy increased CAT enzyme activity and total antioxidant status to similar levels, with their values being 0.18 and 0.17 U/mL, respectively. After the addition of the mimetic, the level of SOD activity was lower at 211.91 U/mL but showed an increase of 49 U/mL compared to the control group (healthy persons).

Analysis of the results obtained for the group of patients after chemoradiotherapy revealed that after the addition of the copper(II) complex, CAT enzyme activity and total TAS approached the level of activity that is characteristic of the control group (healthy subjects). The exception was the activity of the SOD enzyme, whose level in the test group was very high (263.70 ± 126.87) before the addition of the copper(II) compound (Figure 14). Under the influence of the added mimetic, the level of SOD activity in patients after chemoradiotherapy was slightly higher (by 4.2 U/mL) compared to the level of activity in the control group.

In conclusion, studies on the catalytic activity of the investigated compound showed that the copper(II) complex [Cu(2-(HOCH_2_)py)_3_](ClO_4_)_2_ (**1**) could be treated as a functional mimetic of the enzymes tested.

## 3. Experimental

### 3.1. Materials and Instrumentation

Commercially obtained chemicals and solvents were used without further purification. 2-(hydroxymethyl)pyridine (2-(HOCH_2_)py), 2-(hydroxyethyl)pyridine (2-(HOCH_2_CH_2_)py), di(2-pyridyl)ketone (py_2_CO), 2-(2-pyridyl)benzimidazole (pyBIm), copper(II)tetrafluoroborate hydrate, copper(II) perchlorate hexahydrate, 2,2′-azinobis-[3-ethylbenzthiazoline-6-sulfonic acid] and K_2_S_2_O_8_ were obtained from Sigma-Aldrich (Steinheim, Germany) and used as received. Elemental analysis was carried out on Elemental Analyser model VarioMicro Cube (Elementar, Langenselbold, Germany). FTIR spectra were recorded with a Nicolet 380 FT-IR spectrophotometer (Thermo Scientific, Waltham, MA, USA), in the region of 4000–500 cm^−1^ using the diffusive reflection method (ATR); relative intensities are indicated (w: weak, m: medium, s: strong, vs: very strong, br: broad). Electronic spectra of samples dissolved in either methanol or ethanol were recorded on a JASCO V-630 UV-Vis spectrophotometer (Jasco Corporation, Tokyo, Japan) using a quartz cell with a path length of 1 cm. Magnetic moments were measured using a MSB-MK 1 instrument (Sherwood Scientific, Ltd., Cambridge, UK) at ambient temperature with [Hg(Co(SCN)_4_)] as standard. Diamagnetic corrections were carried out using Pascal’s constant. EPR spectra were measured on a Bruker ElexSys E500 instrument (Bruker GmbH, Rheinstetten, Germany) equipped with an NMR teslameter ER 036TM and a frequency counter E 41 FC. The simulation of the experimental spectra were performed using the SPIN computer program [57].

### 3.2. Preparation of Complexes

Perchlorate salts were treated with great caution as they are potentially explosive; they were handled only in small quantities and with care.

#### 3.2.1. Synthesis of [Cu(2-(HOCH_2_)py)_3_](ClO_4_)_2_ (**1**)

Water solution (20 mL) of Cu(ClO_4_)_2_∙ 6H_2_O (0.1 mmol, 0.0370 g) was added dropwise to an ethanolic solution (10 mL) of 2-(HOCH_2_)py (0.2 mmol, 0.0222 g); the mixture was still stirred for 0.5 h at room temperature, and then filtered. After three months, blue crystals of **1** that were suitable for X-ray structure analysis were obtained, collected by filtration, washed with mother liquid and dried in air. Yield 57%. *Anal.* Calc. for CuC_18_H_21_N_3_O_11_Cl_2_ (589.82 g/mol), C, 36.65; H, 3.17; N, 7.12. Found: C, 36.85; H, 3.25; N, 7.12 %. IR (cm^−1^): 3425 (br, w), 1658 (m), 1610 (m), 1576 (w), 1550 (w), 1487 (m), 1394 (w), 1033 (w), 1259 (w), 1167 (w), 1091 (s), 1076 (s), 1068 (s), 1039 (s), 1008 (w), 964 (w), 931 (w), 902 (w), 843 (m), 801 (w), 764 (s), 748 (m), 715 (s), 683 (s), 667 (m), 654 (m), 625 (s), 607 (s), 595 (m), 566 (s), 561 (s), 538 (m), 526 (w), 520 (m), 519 (w).

#### 3.2.2. Synthesis of [Cu(2-(HOCH_2_)py)_2_(H_2_O)_2_]SiF_6_ (**2**)

The reaction of water solution (10 mL) of Cu(BF_4_)_2_∙H_2_O (0.4 mmol, 0.1033 g) with 15 mL of ethanolic solution of 2-(HOCH_2_)py (0.8 mmol, 0.0887 g) resulted in a brilliant blue solution. The mixture was stirred for 1 h at room temperature and left standing for 2 months to give a blue waxy product. Deep green crystals for the structural determination were obtained by diffusion of diethyl ether vapour into an ethanolic solution at r. t. over 3 days. The product was isolated by filtration, washed with mother liquid and dried in vacuum. Yield 23% (based on the copper salts). *Anal.* Calc. for CuC_12_H_18_N_2_O_4_SiF_6_ (459.91 g/mol), C, 31.34; H, 3.94; N, 6.09. Found: C, 31.68; H, 3.21; N, 6.15%;IR (cm^−1^): 3381 (br, m), 2962 (m), 1657 (w), 1649 (w), 1640 (w), 1613 (m), 1573 (w), 1551 (w), 1539 (w), 1511 (w), 1495 (m), 1486 (m), 1448 (m), 1415 (w), 1364 (w), 1347 (w), 1291 (m), 1235 (m), 1189 (w), 1159 (m), 1124 (w), 1107 (w), 1190 (w), 1068 (s), 1054 (s), 1044 (vs), 1034 (s), 1007 (m), 981 (w), 965 (w), 949 (w), 941 (w), 918 (w), 910 (w), 884 (w), 853 (w), 821 (w), 779 (vs), 765 (vs), 725 (vs), 704 (vs), 627 (s), 663 (s), 640 (m), 633 (m), 609 (m), 604 (m), 577 (m), 550 (m), 540 (m), 513 (s).

#### 3.2.3. Synthesis of [Cu_2_(2-(HOCH_2_CH_2_)py)_2_(2-(OCH_2_CH_2_)py)_2_](ClO_4_)_2_ (**3**)

The copper(II) salt solution of Cu(ClO_4_)_2_∙6H_2_O (0.0926 g, 0.25 mmol) in 2 mL of distilled water was dropped into the vigorously stirred ligand solution of 2-hydroxyethylpyridine (2-(HOCH_2_CH_2_)py) (0.1232g, 1 mmol) in 10 mL of ethanol. After 30 min, the colour of the starting solution changed from light green to sea blue. At the end of an hour’s stirring, the solution was filtered and allowed to crystallise. After a week, dark blue crystals appeared, which were filtered under reduced pressure and allowed to dry. Yield 35% (based on the copper salts). *Anal.* Calc. for Cu_2_C_28_H_34_N_4_O_12_Cl_2_ (816.59 g/mol), C, 39.91; H, 4.31; N, 6.65. Found: C, 40.17; H, 3.79; N, 6.72%; IR (cm^−1^): 3450 (br, m), 2858 (w), 2831 (w), 1610 (m), 1570 (w), 1487 (m), 1446 (m), 1437 (w), 1429 (w), 1363 (w), 1340 (w), 1315 (m), 1290 (w), 1250 (w), 1186 (w), 1159 (m), 1086 (s), 1066 (s), 1065 (s), 1051 (s), 1003 (m), 978 (w), 926 (w), 879 (m), 864 (m), 791 (m), 775 (s), 762 (s), 752 (m), 648 (m), 623 (s), 596 (w), 584 (m), 538 (w), 511 (w).

#### 3.2.4. Synthesis of [Cu(pyBIm)_3_](BF_4_)_2_·1.5H_2_O (**4**)

Ethanolic solution (5 mL) of Cu(BF_4_)_2_∙H_2_O (0.05 mmol, 0.0119 g) was added dropwise to an ethanolic solution (10 mL) of 2-(2-pyridyl)benzimidazole (pyBIm) (0.1 mmol, 0.0196 g); the mixture was still stirred for 1 h at room temperature, and then filtered. After 10 days, green crystals of **4** were collected by filtration, washed with mother liquid and dried in air. Yield 64% (based on the copper salts). *Anal.* Calc. for CuC_36_H_30_N_9_O_1.5_B_2_F_8_ (849.83 g/mol), C, 50.88; H, 3.53; N, 14.83. Found: C, 50.49; H, 3.74; N, 14.41%; IR (cm^−1^): 3662 (w), 3592 (w), 2972 (m), 2867 (w), 2670 (w), 1598 (m), 1480 (m), 1450 (s), 1422 (w), 1323 (m), 1293 (m), 1232 (w), 1052 (s), 1015 (s), 938 (w), 911 (w), 820 (w), 798 (w), 748 (s), 695 (m), 621 (w), 564 (s).

#### 3.2.5. Synthesis of [Cu(py_2_C(OH)_2_)_2_](ClO_4_)_2_ (**5**)

An acetonitrile solution (10 mL) of Cu(BF_4_)_2_∙H_2_O (0.25 mmol, 0.0638 g) was added dropwise to an acetonitrile solution (10 mL) of di(2-pyridyl)ketone (py_2_CO) (0.5 mmol, 0.0923 g); the mixture was stirred for 2 h at room temperature, and then filtered. After one week, blue needle-like crystals of **5** that were suitable for X-ray structure analysis were obtained by slow evaporation of the filtrate, collected by filtration, washed with mother liquid and dried in air. Yield 54% (based on the copper salts). *Anal.* Calc. for CuC_22_H_20_N_4_Cl_2_O_12_ (666.86 g/mol), C, 39.62; H, 3.02; N, 8.40. Found: C, 39.55; H, 2.98; N, 8.24%; IR (cm^−1^): 3486 (m), 3297 (m), 3057 (w), 2937 (w), 1657 (m), 1606 (m), 1473 (w), 1447 (m), 1309 (w), 1276 (w), 1236 (m), 1164 (m), 1090 (s), 1031 (m), 1012 (m), 949 (w), 933 (w), 917 (w), 894 (w), 804 (m), 765 (m), 748 (w), 669 (m), 621(s), 570 (w), 543 (w).

### 3.3. Crystallographic Data Collection and Structure Refinement

Diffraction intensity data for a single crystal of copper(II) complexes **2**, **4** and **5** were collected on a KappaCCD (Nonius) diffractometer with graphite-monochromated Mo Kα radiation (λ = 0.71073 Å). Corrections for Lorentz, polarisation and absorption effects [58,59] were applied. The structure was solved by direct methods using the SIR-92 program package [60] and refined using a full-matrix least-square procedure on F^2^ using SHELXL-97 [61]. Anisotropic displacement parameters for all non-hydrogen atoms and isotropic temperature factors for hydrogen atoms were introduced. In the structure, the hydrogen atoms connected to carbon atoms were included in calculated positions from the geometry of molecules, whereas hydrogen atoms of water molecules were included from the difference maps and were refined with isotropic thermal parameters. Single crystal X-ray diffraction data of compounds **1** and **3** were collected on a Stoe IPDS-2T diffractometer with graphite-monochromated Mo-Kαradiation. Data collection and image processing was performed with X-Area 1.75 [62]. Intensity data were scaled with LANA (part of X-Area) in order to minimise differences of intensities of symmetry equivalent reflections (multi-scan method). In either case, crystals were cooled using a Cryostream 800 open flow nitrogen cryostat (Oxford Cryosystems). The structures were solved with direct methods and refined with the SHELX–2016/6 program package [63,64] using the full-matrix least squares procedure based on F2. The Olex [65] and Wingx [66] program suites were used to prepare the final version of CIF files. All C-H type hydrogen atoms were attached at their geometrically expected positions and refined as riding on heavier atoms with the usual constraints. The oxygen atoms of the perchlorate ions in **1** needed to be modelled as disordered: O4–O6 (*s.o.f.* of 0.495(11) and 0.505(11)). The figures were made using DIAMOND software [67]. Crystallographic data for the structures reported in this paper have been deposited with the Cambridge Crystallographic Data Centre as supplementary publications No. CCDC 2079678 (**1**), 2079599 (**2**), 2079677 (**3**), 2079602 (**4**), and 2079600 (**5**). Copies of the data can be obtained free of charge on application to CCDC,12 Union Road, Cambridge CB2 1EZ, UK (Fax: (+44) 1223-336-033; E-mail: deposit@ccdc.cam.ac.uk).

### 3.4. Determination of Antioxidant Activity by the ABTS Test

We investigated the free radical scavenging ability of the selected ligands and their newly synthesised complexes using the ABTS assay. 2,2′-Azinobis-[3-ethylbenzthiazoline-6-sulfonic acid] (ABTS) was dissolved in Milli-Q water to yield a 7 mM solution. The ABTS radical cation solution was prepared by allowing the ABTS solution to react with the K_2_S_2_O_8_ solution (final concentration 2.45 mM) for 16 h in the dark at room temperature. It has been reported [68] that the extinction coefficient of ABTS^•+^ radicals at 734 nm is 1.5 × 10^4^ mol^−1^ L cm^−1^. Using the Beer–Lambert law, ABTS^•+^ radical concentrations were calculated. To determine the antioxidant capacity of the complex solutions and standard antioxidant (Trolox), the ABTS solution was diluted with 5 mM PBS (pH 7.4) to an absorbance at 732 nm of 0.7 ± 0.1 After the addition of different volumes (0–1 mL) of each compound solution to 2.5 mL of diluted ABTS^•+^, the absorbance was measured at 30 min. All samples were analysed in triplicate. The ABTS^•+^radical scavenging activity (AA%) was calculated using the following equation: AA% = ((A_control_ − A_sample_)/(A_control_)) × 100, where A_control_ is the absorbance of the blank control (containing ABTS^•+^ solution without test sample) and A_sample_ is the absorbance of the test sample. The antioxidant activity was described as the 50% inhibition concentration (IC_50_). The IC_50_ values were calculated from regression lines where x is the concentration of complex or ligand in mM and y is the percent inhibition of the analysed compound.

### 3.5. Cell Culture and Viability Assays

For IC_50_ determination, the human cancer lines A549 (non-small cell lung carcinoma), HT29 (colon adenocarcinoma) and MCF-7 (breast adenocarcinoma), and a healthy mouse fibroblast cell line (BALB/3T3), were used. All tested cell lines were obtained from the American Type Culture Collection (Rockville, MD, USA) and maintained at the Cell Culture Collection of the Hirszfeld Institute of Immunology and Experimental Therapy (Wrocław, Poland). The details of the experimental conditions were described in [69]. The results are present as an IC_50_ concentration (μM of tested agent which inhibits proliferation of 50% of cancer cells population). IC_50_ values were calculated separately for each experiment, which was repeated 3–5 times.

### 3.6. The Antioxidant Status of the Plasma in Oncology Patients

#### 3.6.1. Participants, Blood Collection and Processing

The present study was based on 20 women of the Regional Hospital in Kielce who had previously undergone therapy for diagnosed breast tumour, and 10 healthy volunteers (samples were anonymised). Patients were divided into subgroups those after chemotherapy (n = 10) and after radiotherapy (n = 10). The protocol and procedures were undertaken according to the Helsinki Declaration and were approved by the Bioethics Committee of the Świętokrzyska Medical Chamber in Kielce (permission number 1/2015-B). Before obtaining blood, patients and healthy subjects signed informed consent and agreed to the use of the samples for research purposes. Approximately 5 mL of blood was taken from the patients by venous puncture. Blood samples were collected in 3 mL tubes containing EDTA as an anticoagulant, and in tubes with no anticoagulant. Tubes were centrifuged at 3500 rpm for 15 min to obtain plasma and serum, respectively. Erythrocytes were washed with 0.9% sodium chloride. Samples were divided into 0.5 mL aliquots were stored at −80 °C for further analysis.

#### 3.6.2. The Activity of Antioxidant Enzymes

The activities of TAS, SOD, CAT and GPx were measured using commercial kits, according to the manufacturers’ instructions. All assays were performed in triplicates. The concentration of complex **1** in the experiment was 25 μg/mL of blood.

The TAS level was evaluated by RANDOX test (Randox Laboratories, United Kingdom) following the method described by Miller et al. [70]. This method employs ABTS (2,2′-azino-bis (3-ethylbenzothiazoline-6-sulphonic acid)) incubated with a metmyoglobin and H_2_O_2_ to produce the radical cation ABTS^•+^. Suppression of the blue-green colour of radicals (measured at 600 nm) is proportional to antioxidants’ concentration (TAS). The results were expressed as mmol/L serum.

The SOD activity was measured with the RANSOD test (Randox Laboratories, United Kingdom) based on the method of McCord and Fridovich [71]. This method employs xanthine and xanthine oxidase to generate superoxide radicals, which react with 2-(4-iodophenyl)-3-(4-nitrophenol)-5-phenyltetrazolium chloride (INT) to form a red formazan dye. The superoxide dismutase activity is measured by the degree of inhibition of this reaction. One unit of SOD causes a 50% inhibition. The activity of SOD was measured spectrophotometrically at 505 nm and expressed as U/mL.

The CAT activity measurement was measured with the Catalase assay kit (Calbiochem, Merck, Darmstadt, Germany) based on the reaction of the enzyme with methanol in the presence of an optimal concentration of H_2_O_2_ [72]. The formaldehyde produced is measured colourimetrically (at 540 nm) with 4-amino-3-hydrazino- 5-mercapto-1,2,4-triazole (Purpald) as the chromogen.

The activity of GPx was determined using the Glutathione Peroxidase assay kit (Calbiochem, Merck, Darmstadt, Germany) with the method described by Paglia and Valentine [73]. In this assay, GPx catalyses the oxidation of glutathione (GSH) with cumene hydroperoxide. In the presence of glutathione reductase (GR) and NADPH, oxidised glutathione (GSSG) is converted to GSH with concomitant oxidation of NADPH to NADP^+^. The decrease in absorbance was measured at 340 nm at 37 °C (pH 7.2), and the GPx activity was expressed as U/mL.

#### 3.6.3. Statistical Analysis

Results are presented as means ± standard error of the mean (SEM) of samples taken from different individuals. The appropriate comparisons were made for unrelated variables using the Student’s t-test and ANOVA Kruskal–Wallis test, along with the post-hoc Dunn‘s test, utilising STATISTICA software from StatSoft Polska. The level of significance used was *p* < 0.05.

## 4. Conclusions

It is important to note that synthetic antioxidants can support chemo- and radiotherapy for a correct antioxidant balance in organisms. In this regard, we have attempted to synthesise and fully physiochemically characterise five new copper(II) complexes with 2-(hydroxymethyl)pyridine (**1** and **2**), 2-(hydroxyethyl)pyridine (**3**), 2-(2-pyridyl)benzimidazole (**4**) and di(2-pyridyl)ketone (**5**) ligands as potential synthetic mimetics of enzymes. Spectroscopic analyses and X-ray data confirmed that Cu(II) ions in the monomeric complexes possessed distorted octahedral environments with different chromophores: {CuN_3_O_3_} (**1**), {CuN_2_O_4_} (**2**), {CuN_2_O_3_} (**3**), {CuN_6_} (**4**), and {CuN_4_O_2_} (**5**). One of the obtained complexes, in solid state, existed in dimeric form (**3**), in which every metal centre was five-coordinated, forming a distorted square pyramidal geometry. Among the coordination compounds isolated during the syntheses, those soluble in water (**1**, **3** and **5**) were selected as model systems for studies on antioxidant properties. The tested complexes with hydroxymethyl- and ethylpyridine displayed stronger radical scavenging activity (IC_50_ 0.26 ± 0.03 (**1**) and 0.62 ± 0.05 mM (**3**)) than the free ligands (IC_50_ = 3.79 ± 0.23 mM (2-(HOCH_2_)py) and 7 mM (2-(HOCH_2_CH_2_)py)). Complex **5** exhibited the lowest antioxidant activity (IC_50_ = 24.49 ± 0.50 mM). Complex **1** was selected for further studies as a mimetic of the chosen enzymes and tested on the erythrocyte lysate of two groups of patients after chemotherapy and chemoradiotherapy. The obtained data revealed that after the addition of the Cu(II) complex, the level of enzyme activity in chemo-and chemoradiotherapy patients increased compared to controls (healthy patients and who had received chemo-and chemoradiotherapy without Cu(II) compound addition). The effect of the tested compound [Cu(2-(HOCH_2_)py)_3_](ClO_4_)_2_ (**1**) on enzyme activity levels (TAS, SOD and CAT) suggests that it can be treated as a functional mimetic of the enzymes. Future studies should additionally be conducted to evaluate the antioxidant status of patients following chemo- and chemoradiotherapy after a longer recovery period (e.g., 5 years) as a possible indicator of cancer recurrence (e.g., a significantly above average SOD level).

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
