# Peer review of "Synthesis and Structure of Novel Copper(II) Complexes with N,O- or N,N-Donors as Radical Scavengers and a Functional Model of the Active Sites in Metalloenzymes"

_ijms, 2021, doi:10.3390/ijms22147286_

Round 1
Reviewer 1 Report
This manuscript is very well written, hoever before it will be accepted for publication the authors should answer of some questions>
1. What cytotoxic effect possess the new compounds and what toxicity exhibit in comparison to normal cells.
2. The X-ray structures are very important for new complexes but for biological experiment it is important their solubility and stability in water solution. The authors should add this data.
3. For compound 2 i don't see the ml of solution. The authors only showed the amount of moles of compounds used in the reaction.
Reviewer 2 Report
Dear Editor
The manuscript “Synthesis and structure of novel copper(II) complexes with N,O- or N,N-donors as radical scavengers and a functional model of the active sites in metalloenzymes” is a good study in the field of metal complexes and the studying of their biological activity.
I have some questions to the authors which need explanations which are:
1- The formation of the gem diol of the 2-dipyridyl ketone is not logic to me even it is supported by reference, the hemiacetal form of the ketones is stable compound but not the gem diol.
2- The formation of compound 2 (SiF6) derivative in scheme 2 doesn`t include the SiF6 salt as a reactant. The authors stated that the presence of water results this type of complex, what type of water which contains SiO2 (according to the chemical equations written)?
3- All the chemical symbols written (for example: 2-CH2OHpy) are unlogic due to the attachment between the py ring and the CH2OH is the CH2 (and the right for is (2-(HOCH2)py)
4- Table 3 ν(OH)H2O the H2O must corrected.
5- In the schemes show the complexes formation, all bonds shown as covalent bonds which causes a confusion to some readers which Oxygen atom which is trivalent?), I suggest using arrows bonds or dashed bonds.
Regards
